# Regulatory Resistance? Narratives and Uses of Evidence around “Black Market” Provision of Gambling during the British Gambling Act Review

**DOI:** 10.3390/ijerph182111566

**Published:** 2021-11-03

**Authors:** Heather Wardle, Gerda Reith, Fiona Dobbie, Angela Rintoul, Jeremy Shiffman

**Affiliations:** 1School of Social and Political Sciences, University of Glasgow, Glasgow G12 8RT, UK; Gerda.Reith@glasgow.ac.uk; 2Faculty of Public Health and Policy, London School of Hygiene and Tropical Medicine, London WC1H 9SH, UK; 3Usher Institute, Old Medical School, University of Edinburgh, Edinburgh EH8 9AG, UK; fiona.dobbie@ed.ac.uk; 4Health Innovation and Transformation Centre, Federation University, Churchill, VIC 3842, Australia; a.rintoul@federation.edu.au; 5Bloomberg School of Public Health and School of Advanced International Studies, Johns Hopkins University, Baltimore, MD 21205, USA; jeremy.shiffman@jhu.edu

**Keywords:** gambling, Great Britain, Gambling Act review, black market, unhealthy commodities, regulation

## Abstract

Commercial gambling is increasingly viewed as being part of the unhealthy commodities industries, in which products contribute to preventable ill-health globally. Britain has one of the world’s most liberal gambling markets, meaning that the regulatory changes there have implications for developments elsewhere. A review of the British Gambling Act 2005 is underway. This has generated a range of actions by the industry, including mobilising arguments around the threat of the “black market”. We critically explore industry’s framing of these issues as part of their strategy to resist regulatory change during the Gambling Act review. We used a predefined review protocol to explore industry narratives about the “black market” in media reports published between 8 December 2020 and 26 May 2021. Fifty-five articles were identified and reviewed, and themes were narratively synthesised to examine industry framing of the “black market”. The black market was framed in terms of economic threat and loss, and a direct connection was made between its growth and increased regulation. The articles mainly presented gambling industry perspectives uncritically, citing industry-generated evidence (*n* = 40). Industry narratives around the “black market” speak to economically and emotionally salient concerns: fear, safety, consumer freedom and economic growth. This dominant framing in political, mainstream and industry media may influence political and public opinion to support the current status quo: “protecting” the existing regulated market rather than “protecting” people. Debates should be reframed to consider all policy options, especially those designed to protect public health.

## 1. Introduction

### 1.1. Gambling, Regulation and the “Black Market”

Gambling is increasingly recognised as part of the unhealthy commodities industries, contributing to preventable ill-health and mortality [1]. Opportunities for commercial forms of gambling, along with technological advances that change the provision and type of gambling products offered, are growing worldwide. The gambling industry, especially the online industry, is global, with operators providing services and targeting markets across many jurisdictions [2]. The online industry is powering growth in the sector. In Britain alone, the remote gambling sector is now a GBP 5.7 billion industry [3]. Great Britain has one of the most liberal gambling markets in the world, and the ways that it is regulated there has implications for regulatory development elsewhere. Concerns about these developments, particularly its impacts on public health, have been raised [4,5]. This includes recent reports conservatively estimating that the social and economic costs of gambling in England alone amount to at least GBP 1.27 billion [6]. In response to these concerns, in 2020, the UK Government announced a comprehensive review of gambling legislation (the 2005 Gambling Act review), with specific focus on updating legislation for the digital era [7]. This included a focus on potential changes to online gambling provision, including consideration of introducing affordability measures, or deposit and staking limits, as already available in other jurisdictions, such as Sweden or Spain [8].

This review creates uncertainty about how gambling regulation, specifically online gambling regulation, may change. Since 2005, Britain’s gambling laws have been amongst the most liberal in the world, allowing all forms of gambling to be legally provided in land-based venues as well as online. Since the announcement of this legislative review, the gambling industry, along with those advocating for policy change, have engaged in high-profile campaigns to ensure that their views, perspectives and opinions are heard.

Key issues included within the review are the impact of stricter regulation upon the “black market” for gambling. In many jurisdictions, containment of the “black market” especially for online gambling, has a been a driver for the legalisation of online gambling products [9,10]. In Britain, which already has a legal and relatively liberal gambling market, the question is inverted and rather is concerned with the potential impact of greater regulatory restrictions upon demand for alternative “black market” provision.

In terms of international gambling policy, few jurisdictions have imposed greater regulatory restrictions on pre-existing legal and regulated markets. Where this has occurred, there has been little systematic evaluation of the impact. For example, in recent years, one of the most significant changes to British gambling regulation was the 2019 reduction in the maximum stake size from GBP 100 to GBP 2 on so-called Fixed Odd Betting Terminals. As this change was not fully evaluated, we have little insight into the kinds of individual behaviour changes that followed [11]. Elsewhere, Wardle et al. [12] noted that, when gambling opportunities were restricted due to the COVID-19 pandemic, regular gamblers did not appear to switch behaviours or products, and queried the assumptions of linear “black market” transitions. This reflects the experience in Norway, when electronic gambling machines were removed from the market, with Lund [13] concluding that there was no indication of the development of a black market for machines or substitution to other forms of gambling [13]. In general, the evidence base around the impact of regulatory changes on “black market” activity is nascent. This creates a situation in which competing narratives have space and opportunity to fill the void.

As a live policy debate that will continue for the next two to three years, it is essential to understand how issues such as the impact of regulatory change upon the “black market” are framed by key actors, particularly, in the case explored here, the gambling industry. Framing establishes the parameters within which social problems can be discussed and how they are conceived, and it also determines the range of solutions. Social and political scientists contend that debates over framing are power struggles over who defines what counts as a social problem and—importantly—what solutions can solve it. As such, framing is about agenda setting, policy formation and resource allocation [14,15,16,17]. Reflecting this, researchers [1,18] advocated for taking a systematic approach to understanding the actions of the unhealthy commodities industries, including gambling, in an attempt to restrain public health action. Knai et al. [1] argued for the need to “focus on the narratives constructed by political actors attempting to frame political problems, attribute responsibility for them and advocate particular solutions”. However, a systematic, empirical, evidence base that examines these issues within a British context is only just starting to emerge [19].

The announcement of the Gambling Act review and the specific questions it posed around the “black market” created a space for the gambling industry to put forward its opinions about the impact of regulation. The narrative advanced here, in common with those of other unhealthy commodity industries, such as tobacco and alcohol, is based on the argument that increasing regulation increases the “black market” for a commodity [16,20]. However, unlike physical products such as tobacco or alcohol, gambling is increasingly a digital commodity. In this context, concerns about containment and control involved in “black market” narratives are particularly salient.

Focus on so-called ”black markets” is part of a wider industry “playbook” whereby companies deploy strategies to resist regulation and to undermine public health initiatives. These include political lobbying; research sponsorship; and promoting discourses that legitimise their business model, including casting doubt on scientific evidence and mobilising arguments about complexity [21,22,23,24]. They also include mobilising narratives around the potential threat of unlicensed markets: a tactic the tobacco industry has frequently employed [25,26]. Gallagher et al. [20] noted that transnational tobacco companies (TTCs) frequently fund (unreliable) research into illicit trade and use the reports to back claims about the threat of the black market and to resist policies of tobacco control. As van Schalkwyk et al. [19] highlighted, focus on illicit trade is a familiar trope. When applied to gambling, the potential reach and reception of these framings are heightened not only because of the increasingly digital and transnational nature of the product but also because of a lack of credible and independent evidence about the scale of the “black market” itself [27]. This generates the conditions whereby industry framing of these issues speak to a number of economically and emotionally salient concerns, namely fear, safety, consumer freedom and economic growth.

Examining the ways in which the gambling industry frame debates about the “black market” provides a case study of how they attempt to influence policy action and to enact forms of regulatory resistance. In this paper, we explore this by conducting a narrative content analysis of media articles to examine how arguments around the so-called “black market” are mobilised and who is mobilising them. In doing so, we critically examine what messages are communicated via different mediums and what framing devices are used to shape the debate and assess how these may have changed. First, we present some background on the regulatory and policy context of gambling in Britain. We outline the background of the Gambling Act review before outlining the actions taken by key representatives of the gambling industry in response to it.

### 1.2. The Gambling Act Review and the “Black Market”

A review of the Gambling Act 2005 was a 2019 General Election manifesto promise for most major political parties in Great Britain, including the now incumbent ruling party, the Conservatives. In February 2020, the National Audit Office [28] released its assessment of the performance of the Gambling Commission, the industry regulator, highlighting structural issues with how gambling is regulated. In July 2020, the House of Lords Select Committee Enquiry [29] released the findings of its inquiry on the social and economic impacts of the gambling industry, arguing that it was time for change. In addition, an All Party Parliamentary Group tasked with examining gambling harms has been increasingly vocal about the need for changes in the way gambling is regulated, promoted and provided in Britain. The official review was launched on 8 December 2020 [7]. It opened with a call for evidence on over 40 questions of interest. Three questions were specifically related to potential “black market” activity:What, if any, evidence is there to suggest that there is currently a significant black market for gambling in Great Britain or that there is a risk of one emerging?What evidence, if any, is there on the ease with which consumers can access black market gambling websites in Great Britain?How easy is it for consumers to tell that they are using an unlicensed illegal operator?

The call for evidence closed on 31 March 2021. Responses to these questions will form the basis of a further public consultation on a draft white paper outlining suggested changes, likely to be launched in late 2021/early 2022. Thus, at the time of writing, Great Britain is in the early stages of the Gambling Act review process. Its formal announcement generated a significant amount of research and promotional activity by all stakeholders.

Although used in the review, the term ”black market” covers a range of gambling products and services that are not legally offered in Britain. Britain provides a licensed gambling market requiring operators to seek permission to operate from the Gambling Commission (GC). Technically, anyone providing gambling services to Great Britain without a valid license is illegally providing gambling services in Britain. Unlicensed actors may range from corporations licensed in other jurisdictions to criminal syndicates [30]. Some unlicensed operators may be licensed and regulated elsewhere, whereas others are not. In addition, the illegal market for gambling includes gambling and betting with digital items, such as skins. In short, there is considerable linguistic ambiguity about the spectrum of actors that constitute the “black market” in gambling.

### 1.3. Initial Gambling Industry Responses to the Gambling Act Review

The main leaders in lobbying for the British gambling industry are the newly formed Betting and Gaming Council (BGC), who are now the single trade body for the British gambling industry. The BGC focuses on promoting so-called “responsible gambling”, and their aims and objectives are, in their terms, to “safeguard” and “empower” the customer as the key to a thriving UK gaming and betting industry and to ensure that changes in regulation are, in their terms, “considered, proportionate and balanced”. On 20 December 2020, 12 days after the formal announcement of the Gambling Act review, the BGC began a campaign to promote its position and perspective on the “black market”. To begin, they issued a press release entitled “Black Market Threat”, quoting unpublished research conducted by PriceWaterhouseCoopers (PWC, London, UK) in 2018/19. This was followed by a new report, also authored by PWC, published on 4 February 2021 [31], accompanied by a media strategy to disseminate the findings. This included personal appearances from BGC spokespeople on radio and television media, interviews with newspapers, and opinion pieces in newspapers such as *The Times* [32]. These actions sparked a wave of further commentary, including from the industry regulator (the GC) and others within the gambling industry as well as from academic circles. A chronology of these events is illustrated in Figure 1, along with other key publications during this period.

To explore these actions further, we conducted a narrative media analysis to review and critically examine the early arguments made around questions of the “black market”. Our analysis was guided by the following research questions:What is the nature of reporting and discussion around unlicensed gambling in the UK in the initial stages of the Gambling Act review (December 2020–May 2021)?What type of publications/authors invoke the use of the “black market”, and how is this framed?What is omitted/not discussed from this public discussion?

## 2. Materials and Methods

To analyse messaging and reporting around themes of the “black market”, we conducted a narrative media content review using Google News and LexisNexis (New York, NY, USA) to search for all media reports discussing the “black”, “unregulated”, “unlicensed” or “illegal” market in Britain between 8 December and 27 May 2021. A review protocol was produced and agreed upon by all team members (see the Appendix A). Searches were conducted by H.W. and F.D. in the week commencing on 24 May 2021 (see Figure 2). After screening for eligibility, all relevant articles were extracted and summarised by type of article, type of publication, whether the “black market” was the key focus, who broadly discussed the “black market”, whether the “black market” was mentioned in response to other research data or regulatory change, and whether any critiques of the “black market” analysis or arguments were included. Articles were excluded if they were not UK-focused, did not mention the “black market” for gambling or were outside the data range. Data extraction was undertaken by H.W. and checked by F.D. The articles were read and reread by H.W. to initially identify the broad narrative themes within each article [33]. These were discussed with the broader team, and the thematic groupings were agreed upon. The data extraction framework was amended so that information on each theme within the analysis could be recorded (where possible) for each article. This included recording who discussed the “black market”, what the broader context of the article was and whether any countervailing viewpoints were presented. The first ten articles were extracted using this framework, and the themes were discussed and finalised with the authors. Once all extraction was completed, a final narrative synthesis of themes was undertaken. The articles were also coded to examine their tone, in terms of whether they were pro-industry, balanced or industry critical. Coding was guided by noting whether only one perspective was presented, whether only industry sources were quoted, and whether any critical discussion of industry perspectives or countervailing viewpoints were discussed. This coding was undertaken independently by two raters, and the results were compared (inter-relater reliability was k = 0.71, defined as substantial agreement according to Landis and Koch [34]). Disagreements were discussed between the two coders, and a final assessment was agreed upon. Finally, to further assess heterogeneity within the review, the articles were also coded for their source (industry press, national press, regional press and political press). All data were recorded in an excel database, allowing analysts to filter by article type to assess whether the results varied across different types of media publications (database available upon request).

## 3. Results

### 3.1. Search Results and Characteristics of Studies

Once duplicates were removed, 55 media articles were included within the review (Figure 2 and Table 1 for the articles included in the review). This included 33 articles from industry press, 12 from national press, 6 from regional press and 4 from political press. The majority were news reports, reporting the latest developments impacting the gambling industry and/or policies (*n* = 31). The remainder were opinion pieces or review articles, giving an overview of key issues around the Gambling Act review. By-lines were attributed to 28 different journalists, with the BGC having four articles where the by-line was attributed to them. Most articles presented one perspective only on the risk of the black market (*n* = 40), namely that of the industry, and the tone of most articles reflected industry viewpoints uncritically, with little countervailing views or opinions presented (*n* = 36). Only three articles were designated as industry critical (all published in national mainstream media), whilst the remaining articles presented more balanced perspectives from a range of stakeholders (*n* = 16). Whilst this may seem unsurprising, given the bias towards industry-focused publications included within the review, not all pro-industry articles came from this source. Of those from national, regional or political press (*n* = 22); 13 were categorised as being pro-industry in tone.

### 3.2. Narrative Themes Identified within Articles

Our analysis identified six interlinked themes that typify how media have discussed the “black market” for gambling since the launch of the Gambling Act review. These themes focused on “black market” growth, determinism and displacement, economic and political contribution and loss, safety and protection, and retort and response. Each of these are discussed in detail below.

#### 3.2.1. “Black Market” Growth

Many articles cited the reports issued by the BGC (*n* = 27 out of 55), conducted by the consultancy firm PWC, which claims that the “black market” is currently a growing threat. Citing statistics generated by an online survey, headlines focused on the black market “doubling” in size between 2018/19 and 2020. This was based on the report finding that the proportion of online gamblers who said that they had used an unlicensed operator grew from 2.2% in 2018/19 to 4.5% in November 2020 [31]. This statistic was reported in the industry, national and regional press. This narrative was repeated by industry spokespeople in subsequent reports, describing the “growing online black market” [74,75,83,87] or criticising those who questioned these data by saying:
“There are people in the anti-gambling lobbying regimes that seem to deny that the black market exists…To me that is sort of saying that the Earth is flat. It’s very clear that there is a black market for gambling and I think anyone should be worried by the fact that the numbers have doubled in about a year”[61].

A few media articles carried critiques of the data from the PWC reports. They tended to focus on the Gambling Commission’s prior rebuttal that data from these reports were “not consistent with the intelligence picture” [35,43] and highlighted flaws in the study methodology. A further critique of the survey results was published by Howard Reed in the magazine *Left Foot Forward* [62]. Reed suggested that the apparent increase between years could be an artefact of the COVID-19 pandemic, with people substituting terrestrial for online gambling, and noted that the same report shows a marked decline in “black market” visits when measured using Google Analytics [62]. This contextual finding was omitted from many media reports. For Reed, the critical point was that the current “black market” share is relatively small, and he advocated focusing on the performance of the regulated market. Others agreed, stating that more was needed to understand why and how people were gambling outside of the regulated markets:
“[T]he biggest single challenge in making any real impact assessment of any black market is that no-one knows the real scale or scope of such a market, by definition. It is not measured, not is it easily measurable.”[27].

However, these critiques and rebuttals were only evident in a minority of articles (*n* = 15). Most simply carried the narrative of growth with no contextualisation or critical analysis. 

#### 3.2.2. Determinism and Displacement

The relationship between regulatory change and risk of an emerging “black market” was presented in a deterministic fashion in many media reports. This tended to follow the argument that, if you restrict gambling in Britain, people will seek these products elsewhere. Media reports that included this theme tended to directly quote industry spokespeople on this point:
“If people were restricted, they would simply migrate to the growing unlicensed, unsafe, black market”[83].
“If checks are unnecessarily onerous, punters will go to the unsafe black market”[36].
“Black market betting will rocket if new laws are too strict”[70].

These remarks from industry commentators suggest a wholesale shift from regulated to unlicensed markets in the face of further regulatory restrictions. The potential restrictions that the industry responded to are varied, ranging from affordability checks—which involves checks on the ability of gamblers to pay—to online product control—such as reduced stake sizes. These public health measures were described as “draconian” [79], “onerous” [47], interventions from the “nanny state” and “the forces of prohibition” [64], which would result in “over-regulation” [54]. The implication was that further regulatory intervention would accelerate the threat of the “black market”. A few of the news reports (*n* = 10 out of 39) that included such industry statements critiqued or contextualised these statements. Contextualisation tended to be provided within opinion pieces, with one author noting the following:
“The risks of driving people to the black market may be offset by more expansive powers and funding for the Gambling Commission”[81].

Others cited the riposte issued by the Gambling Commission that the industry was exaggerating the threat of the black market [42,43,48,54,56,57,66,71].

#### 3.2.3. Economic and Political Contribution and Loss

When discussing the “black market”, some articles invoked a parallel narrative based on the idea that gambling contributes to the British economy; thus, any movement of gambling to the illegal market represents a loss for the UK economy overall. These statements, put forward by industry spokespeople, tended to be coupled with “warnings” to policy makers and government to think carefully before taking action that would risk economic contribution through taxes and employment:
“The regulated betting and gaming industry employs 100,000 men and women and pays £3.2 bn a year in tax to the Treasury, so the Government needs to be wary of doing anything that puts that at risk.”[40].

The “black market”, by contrast, was typified as follows:
“[U]nlicensed, unsafe [black market online] that employs no one, pays no tax and contributes nothing to UK PLC”[74,83,87].

Loss was not only typified as economic loss but also as potential loss of political support. In one article, the BGC noted the following:
“I hope politicians will also take heed of the findings and listen to voters in Northern and Midlands marginal seats—who will be key to the result of the next election—who are wary of being told by Westminster how to live their lives, especially in the wake of the COVID pandemic”[71].

#### 3.2.4. Safety and Protection

The term “unsafe” was commonly used when describing the “black market”, specifically referring to the lack of consumer protection offered by “black market” operators. Citing statements from industry representatives, some articles stated that “black market” operators had “no consumer protections” [35,36]. Others went further and described the black market as “dangerous” [54,64,67]. Parallels were drawn with organised crime, and one article included a quote describing “black market” online operators as follows:
“[The] modern day internet equivalent of the ‘Peaky Blinders’—dangerous, illegal backstreet bookies, run by organised crime”[70].

A BGC representative emphasised the “unsafe” nature of the “black market” to certain individuals by stating the following:
“I have never met a problem gambler who has not gambled on the black market”[50].

As well as linking illegal gambling to organised crime, it was common to emphasise that unlicensed operators have none of the consumer protection safeguards that currently exist within the British, licensed marketplace and that this was a threat to consumer safety. Here, narratives of consumer safety intersected with lost revenues, with the argument put forward that once people are lost to the “black market”, the regulated market can no longer protect them [38,52,54,67].

#### 3.2.5. Balance and Proportion

A common theme in the discussion around the “black market” was an appeal to balance—that changes in regulation should be balanced against the threat of the growing “black market”. Some also discussed achieving the correct balance between protecting what they termed as the few or “tiny minority” [79] harmed against infringing the liberty of the many [36]. This tended to be discussed in terms of narratives of proportionality, reasonableness and care, emphasising cautious approaches to regulatory change from the perspective of the industry:
“It is important to stress that the big increase in the black market is not an argument against more changes to the regulated industry, but an argument that we need to get them right” [54].
“It is important the review is evidence-led, strikes the right balance between protecting the vulnerable and the continued enjoyment of the many tens of millions who happily place the occasional bet, as well as taking a critical look at the growing risks of the black market”[35].
“You’ve got to be careful and reasonable in what you do. A £2 limit on table games [such as online roulette] would give you a problem with the black market.”[52].

#### 3.2.6. Retort and Response

Finally, in some articles, the “black market” threat was used as a retort to publications that presented information supporting regulatory change. For example, the BGC responded to two independent reports on the economic impact of gambling that suggested that regulatory restrictions would have a net benefit on the economy overall by re-emphasising the narrative that restrictions would move people to the “black market” [84]. In response to a report published by NatCen Social Research (London, UK), which found that 70% of revenue from online casinos was generated from 5% of players, industry responses emphasised the steps they had taken to raise standards and how none of these standards applied to the unsafe “black market” [75,77]. Industry announcements of support for “safer” gambling weeks were accompanied by statements that none of the measures in the regulated market apply to the unregulated market [82]. The narrative of the “black market” was mobilised in industry responses to other initiatives: when showing support for the Gambling Commission’s introduction of safer game standards, the BGC noted that this did not exist for “black market” operators [53]. In an article covering the “Big Step” movement—a charity that raises awareness of gambling harms and seeks to end the relationship between football and gambling—an industry representative stated that sponsorship and advertising were a vital way for consumers to distinguish between the legal and illegal market [69].

## 4. Discussion

Since the announcement of the Gambling Act review in December 2020, debates about the “black market” have been increasingly weaponised by certain sections of the gambling industry. This trend was exacerbated by the inclusion of specific questions on the issue of the black market within the Gambling Act review’s call for evidence. The gambling industry, similar to other UCIs, utilises a number of strategies to avoid regulatory action, to undermine public health reform and to promote their agenda. Framing is a crucial part of this agenda [1]. It establishes the parameters for debate, defining how we think about an issue and—crucially—about what we can do about it.

Our analysis shows some of the ways in which industry actors have sought to promote a particular framing of the issue of the “black market” in an attempt to influence public and political debate around tighter regulation using a framing that exploits a number of politically, economically and emotionally salient concerns. These relate to fear, safety, consumer freedom and economic growth.

These industry narratives are embedded within a deterministic causal framework based on the proposition that, if the government enacts greater regulatory restrictions, there will be a wholesale shift of consumers to the “unsafe” black market. This framing has economic and emotional connotations, positioning a potential regulatory shift as representing a loss of contribution to the economy and loss of opportunity to protect consumers from harms. The question about the impact of regulatory change and its relationship to the “black market” is an important one. However, it requires a more nuanced assessment of the shape and nature of the relationship than appears evident in the articles reviewed. This relates both to the type of restrictions proposed, the number of people these restrictions are likely to effect and the range of mitigating actions that could be implemented to restrain the “black market”. For example, most of the restrictions suggested pertain only to online gambling, an activity in which only around ten percent of the population engage, with even fewer engaging very frequently [88]. Some have suggested that the value of deposit limits be set at around GBP 100 per month, the median amount spent by online gamblers [89]. Thus, the number of people likely affected by these changes are a minority of a minority. Whilst this minority is clearly lucrative for the industry, the industry itself would also be heavily invested in limiting “black market” movement, and it is likely that they would establish and support systems to help detect and prevent this. Very few of the articles reviewed included this kind of nuance, instead presenting arguments as a dichotomy between prohibition and legalisation and the size of the “black market” being inversely related to each. This is a relationship that recent reports around the legalisation of marijuana in North America, for example, have casts doubt upon in terms of noting potential black market expansion in the face of legalisation [90].

Notably, the assumption of a linear relationship between greater restrictions and migration to “black market” trade is not backed by evidence. The BGC’s own research showed a significant reduction in unique unlicensed operator sites between 2018/19 and 2020 [31], despite that being a period of increased regulatory strictures (for example, removing the use of credit cards for gambling). The regulator themselves stated that notions of black market growth were inconsistent with their own intelligence picture [43]. Experience from other UCI’s, especially tobacco, suggests that industry claims of increases in illicit trade as a result of greater regulation should be treated with caution [20], specifically when results from industry-funded research is compared with independent data [91]. Indeed, based on critical analyses of industry actions within tobacco, researchers have concluded that industry estimates of the impact of regulatory change on “black market provision” are over-stated [20,69]. However, similar evidence and investigation tends to be lacking for the field of gambling. As Richardson states (quoted in [27]) independent and credible research into the scale and scope of the black market in gambling is nascent. This creates a lacuna in the evidence base into which industry-funded research can move. This is clearly the case in Britain, where the BGC produces press releases of reports that are not available to the public and publicises subsequent research that has not been subjected to peer review. At the same time, the regulator’s own intelligence, suggesting that these reports over-estimate concerns, are downplayed by the industry. Evidence from other UCI’s, such as tobacco, show similarities between the actions of the tobacco industry around their assessment of the risks of regulatory change upon the “black market” and tactics currently employed by some elements of the British Gambling Industry [20,91].

Industry framing of the “black market” threat is situated within broader debates about the role of gambling in British life and culture. These include claims about the supposed economic contribution of the gambling industry in terms of the creation of jobs and tax contribution. They are entwined with appeals to the idea of national wealth and counterposed with the idea of its loss to foreign competition: a trope that has a long historical lineage in attempts to justify protectionist economic policies [92]. Here, industry actors argue that the current “positive” contribution to the “UK PLC”, that is, the UK economy, must be maintained and that tax revenue is not lost to foreign entities. These assertions have been critically examined by two independent reports, both of which state that the contribution gambling makes to the economy is over-stated and that economic benefits may be accrued by restricting gambling. This occurs because consumer expenditure would be diverted elsewhere, towards commodities with greater economic multiplier effects, so generating greater, not less, economic contributions. Neither of these reports consider the societal costs engendered by gambling harms, which recently have been estimated to be in excess of GBP 1.27 billion [6]. However, responses to these economic research reports, which were published during the review period, were dismissed by the industry as a “fantasy”, and the threat of the “black market” was reiterated [83].

Themes that run throughout our analysis include notions of “responsible gambling” and “freedom of choice” [93,94]. These, in turn, are embedded in broader neoliberal ideologies about “free” consumer choice and loosely regulated markets. Industry-based narratives that claim the “enjoyment” of millions of gamblers are threatened by onerous regulatory change seeking to downplay the number of people harmed by gambling and focusing on ideas about recreation, leisure and choice of the “responsible majority” instead [94]. This kind of framing is located within wider neoliberal tropes of consumer sovereignty and a rejection of what is framed as “paternalistic” state interference [92,95]. It is also employed by other UCIs, who claim that the freedom to choose is sacrosanct and that choice should not be constrained by the state—typically referred to in these kinds of narratives as the ”nanny state”—even in the pursuit of health protection [21,96].

Our analysis revealed little resistance to these industry-led narratives. Few of the articles reviewed attempted to critically examine the circumstances and actions that may restrain or protect against “black market” activity. Only 15 out of the 55 articles reviewed included any kind of countervailing perspectives to industry narratives. A more nuanced assessment of the risks and solutions necessary to restrain further growth of illegal gambling activity are, by and large, absent from the framing of these issues. Other jurisdictions have instituted disruption measures to reduce unlicensed gambling, including pop-up messaging warning of accessing unlawful websites, working with banking sectors to block unlicensed merchant codes and international collaboration between regulators. An industry poll showed that 73% of regular British gamblers were concerned about privacy rights and data sharing [79]. Given this, campaigns that highlight how data security risks are exacerbated within the unlicensed market could deter engagement. Only one article noted that the Gambling Commission’s resources could be increased to mitigate the “black market” risk, a notion not without merit. In 2019, the gambling industry praised the actions of the Gambling Commission:
“The UK has been highly successful in suppressing the development of a black market for gambling. This is likely to be the result of laws to permit the range of betting and gaming products that consumers wish to play; the maintenance of taxes (with some exceptions) at reasonably sensible levels; and by effective enforcement by the Gambling Commission and other relevant authorities” [97].

A regulator with expanded resource and capabilities could continue to undertake effective enforcement action. Whilst some articles did contextualise industry statements on the “black market”, typically by referring to the Gambling Commission’s statement that the industry may be exaggerating these data, most did not. It was not only industry press presenting only one viewpoint but also articles published in mainstream and national media, including *The Sun*, *The Guardian* and *The Telegraph*.

To date, the dominant narrative surrounding the “black market” is framed simply as one of threat—to consumers, to the economy and to the regulated industry—in what appears to be a subversion of the precautionary principle, whereby regulatory actions should be curtailed in order to protect the interests of the existing licensed gambling industry [98]. This was apparent in 45 of the 55 articles reviewed, with most articles simply reflecting industry perspectives, with no countervailing viewpoints presented or discussed. This is, perhaps, to be expected; there was a dominance of industry press within the articles reviewed and some were opinion pieces written by industry representatives. However, even within mainstream press, a bias towards simply reflecting industry narratives was evident; 9 out of 17 mainstream news articles were pro-industry in tone and reporting. This highlights concerning and potentially systematic biases in how media are reporting, co-constructing and legitimising industry narratives through the absence of detailed critic or analysis of the insight, evidence and arguments put forward.

We systematically extracted and reviewed all articles from two databases in accordance with a pre-specified review protocol. Whilst every effort was made to include all relevant media articles that were published between 8 December and 26 May, we did not have access to articles behind paywalls (usually specialist industry press) and this may have limited the quantity of the studies included. However, of the studies reviewed, the same themes were evident across them, particularly as many focused on simply reporting the details from industry press releases, with the same quotes being cited in many reports.

## 5. Conclusions

Successful framing is a political act. It determines who sets the agenda for how we think about gambling and how we deal with it, as well as precludes serious discussion of alternatives. Industry framing around the potential threat of a “black market” during the period in which the Gambling Act was under review is a form of regulatory resistance. It can be seen as part of a wider industry “playbook” designed to undermine public health initiatives.

The evidence base on which arguments about the “black market” threat is advanced is, at best, limited and, at worst, unreliable. This creates a space into which industry-funded narratives can move and fill with their own set of claims and opinions. There is a dearth of independent, empirical evidence with which to assess the veracity of these claims. This situation is arguably exacerbated by systematic under-funding of gambling research in Britain, especially in situations where regulatory changes have been implemented but not evaluated [11,99]. Despite this, through concerted public relations efforts and with the (knowing or unknowing) complicitly of some media outlets, a narrative around the supposed threat of a “black market” has emerged and gained traction. By framing the debate at an early stage of the Gambling Act review process, this has the potential to influence both political and public opinion in ways that support the current status quo in favour of industry-friendly actions to “protect” the “legitimate” market and to “safeguard” consumers. Such a framing potentially precludes alternative, more far-reaching policy solutions that are framed in terms of the protection of the population from harms. While the latter is informed by concerns about the impacts of commercial practices on the health and wellbeing of the public, policymaking that is framed in terms of concerns about the “threat” of unlicenced trade is informed by a focus on the financial health of commercial operators. These different framings both support as well as preclude very different policy options. Policies in which gambling is framed as a potential source of harm are able to consider actions such as greater regulation and a restriction of commercial practices such as advertising and promotion. However, when gambling is framed only in terms of economic growth, the available options involve actions, such as reduced regulation, to support commercial practices and to “protect” the industry from the “black market”.

The terms of the debate need to be reframed in a way that allows for consideration of all of the policy options available, particularly those that are designed to protect the health and wellbeing of the public from the threat of gambling harms and not merely from the potential “threat” of a black market. To support these debates, understanding of the “black market” in Britain needs to be supported by credible evidence based on independent and rigorous research.

## Figures and Tables

**Figure 1 ijerph-18-11566-f001:**
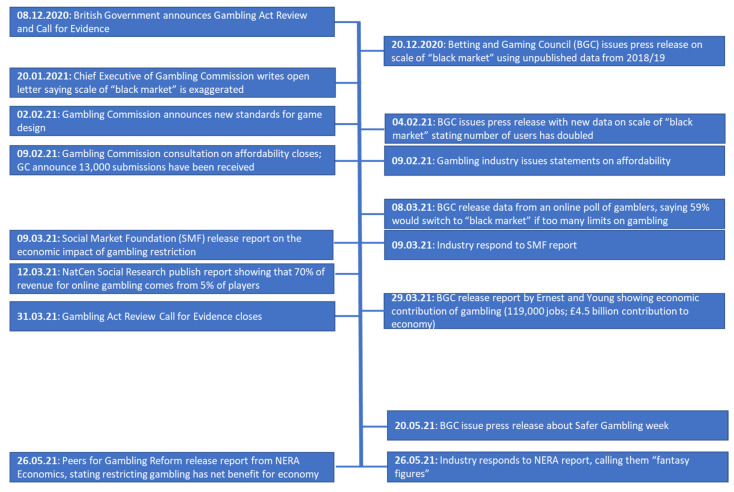
Timeline of key events.

**Figure 2 ijerph-18-11566-f002:**
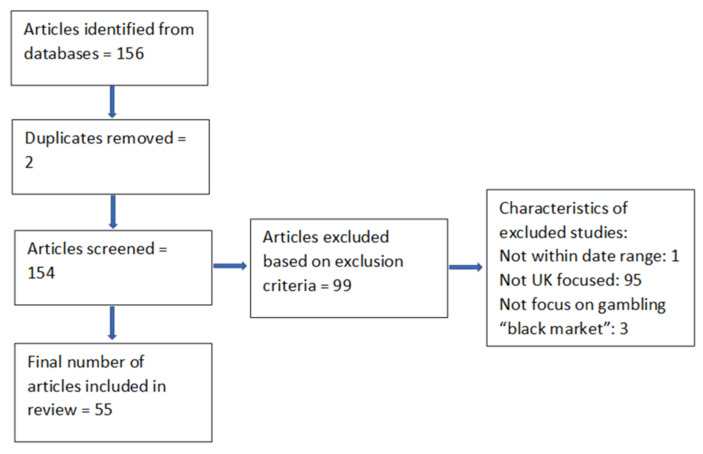
Search process and results.

**Table 1 ijerph-18-11566-t001:** Summary of articles included in the review.

Title	Publication	Publication Type	Date	By-Line	Article Type	Tone
William Hill CEO: important that gambling review strikes right balance [35]	*Gambling Insider*	Industry press	8 December 2021	Tim Poole	Opinion piece	Pro-industry
Betting and Gaming Council chair cautions against severe gambling review [36]	*Racing Post*	Industry press	8 December 2021	Jonathan Harding	News piece	Pro-industry
UK government launches consultation on updating gambling laws for digital age [37]	*Mondaq*	Legal press	10 December 2021	Geraint Lloyd-Taylor	Opinion piece	Balanced
Betting and Gaming Council issues renewed black market warning [38]	*Casino Beats*	Industry press	21 December 2020	NA	News report	Pro-industry
UK sees 27 m visits to black market sites in a year [39]	*iGaming Business*	Industry press	21 December 2020	Richard Mulligan	News report	Pro-industry
UK online black market sees 27 m visits in a year: BGC warns over gambling review [40]	*Yogonet*	Industry press	22 December 2020	NA	News report	Pro-industry
Government must tackle rates gambling addiction which crisis waiting happen according to MP [41]	*Dewsbury Reporter*	Regional press	7 January 2021	Ian Hirst	News report	Balanced
UK gambling operators criticized for overstating black market threat [42]	*Gambling News*	Industry press	18 January 2021	Fiona Simmons	News report	Balanced
UK gambling firms accused of exaggerating scale of black market betting [43]	*The Guardian*	National press	18 January 2021	Rob Davies	News report	Industry-critical
Teens blowing 1.4 billion black market gambling [44]	*The Sun*	National press	20 January 2021	David Wooding	News report	Pro-industry
Winning Post playing it straight on black market threats [45]	*SBC News*	Industry press	25 January 2021	Regulus Partners	Opinion piece	Pro-industry
Sunak monitors gambling review closely [46]	*SBC News*	Industry press	29 January 2021	Ted Menmuir	News report	Pro-industry
Rishi Sunak raises concerns of enhanced affordability checks [47]	*Inkedin*	Industry press	29 January 2021	NA	News report	
Higher license fees to help gambling commission address modern challenges [48]	*Gambling News*	Industry press	31 January 2021	Fiona Simmons	News report	Balanced
Cruddace shares concerns on gambling review [49]	*Racing TV*	Industry press	31 January 2021	NA	News report	Balanced
Cruddace airs concerns on gambling review [50]	*Evening Times* (Glasgow)	Regional press	01 February 2021	Matthew Johnstone	News report	Balanced
Entain launches consumer representation initiative the players panel [51]	*SBC News*	Industry press	01 February 2021	Ted Orme-clay	News report	Pro-industry
UK betting faces bigger threats than losing its sport shirts [52]	*The Guardian*	National press	01 February 2021	Rob Davies	Opinion piece	Balanced
BGC determined to drive change as UKGC clamps down on slots [53]	*Casino beats*	Industry press	02 February 2021	NA	News report	Pro-industry
Industry urges GC to heed black market warnings in gambling act review [54]	*iGaming Business*	Industry press	04 February 2021	Robert Fletcher	News report	Pro-industry
Number of British punters using black market gambling sites more than doubles [55]	*Politics Home*	Political press	04. February 2020	BGC	Opinion piece	Pro-industry
Black market doubles during pandemic [56]	*The Telegraph*	National press	04 February 2021	Oliver Gill	News report	Pro-industry
Even low levels of gambling linked to financial hardship study finds [57]	*The Guardian*	National press	04 February 2021	Rob Davies	News report	Balanced
Ministers should beware driving gamblers to the black market [32]	*The Times*	National press	04 February 2021	Michael Dugher, BGC	Opinion piece	Pro-industry
Gambling limits an invasion of privacy [58]	*Sunday Express*	National press	07 February 2021	David Maddox	News report	Pro-industry
PWC report shows British black market gambling doubled since 2018 [59]	*City A.M.*	National press	05 February 2021	Clara Dijstrak	News report	Pro-industry
Its long overdue: Northern Ireland gambling laws set for significant reform [60]	*Racing Post*	Industry press	08 February 2021	Bill Barber	News report	Pro-industry
William Hill’s Ulrik Bengtsson: claiming the black market doesn’t exist is like saying the earth is flat [61]	*SBC News*	Industry press	08 February 2021	NA	Opinion piece	Pro-industry
Bad bet: how the gambling industry is lobbying hard against online betting reform [62]	*Left Foot Forward*	Political press	09 February 2021	Howard Reed	Opinion piece	Industry-critical
BLACK MARKET DANGER MP warns of ‘disastrous’ consequences affordability checks on punters will have on racing [63]	*The Sun*	National press	09 February 2021	Bruce Archer	News report	Pro-industry
Donoughue cuts ties with all party betting gaming group [64]	*SBC News*	Industry press	12 February 2021	Ted Menmuir	News report	Pro-industry
Flutter’s Ian Proctor outlines affordability approach [65]	*SBC News*	Industry press	12 February 2021	Ted Orme-clay	News report	Pro-industry
British Racing bodies hit out at disastrous affordability checks plans [66]	*iGaming Business*	Industry press	15 February 2021	Robert Fletcher	News report	Pro-industry
Government’s review into gambling laws must take heed of black market threat warn industry experts [67]	*London Post*	Regional press	15 February 2021	NA	News report	Pro-industry
All bets are off for gambling reforms in the UK [68]	*London on the inside*	Regional press	25 February 2021	NA	Review	Balanced
Stadium walks held to call for end to football betting ads [69]	*BBC News*	National press	06 March 2021	NA	News report	Industry critical
ILLEGAL BET SITE FEARS Black market betting ‘will rocket if new laws are too strict’ [70]	*The Sun*	National press	07 March 2021	David Wooding	News report	Pro-industry
BGC poll reveals public opposition to bet limits [71]	*SBC News*	Industry press	8 March 2021	Ted Orme-clay	News report	Pro-industry
Poll: Gambling Limits Prove Unpopular in Britain [72]	*The Blood-Horse*	Industry press	08 March 2021	Scott Burton	News report	Pro-industry
Brits oppose gambling limits poll concludes [73]	*Casino.org*	Industry press	09 March 2021	Devin O’Connor	News report	Pro-industry
SMF: UK economy could benefit from strict gambling laws [74]	*Gambling Insider*	Industry press	10 March 2021	NA	News report	Balanced
Study claims 5% of accounts responsible for 70% of British GGY [75]	*iGaming Business*	Industry press	12 March 2021	Daniel O’Boyle	News report	Pro-industry
Consultant: More research into black market motivations needed instead of “misplaced critique” [27]	*Gambling Insider*	Industry press	12 March 2021	Iqbal Jonal	Opinion piece	Balanced
BGC reasserts zero tolerance approach to betting by under-18 s [76]	*Casino Beats*	Industry press	15 March 2021	NA	News report	Pro-industry
Natcen report fails to highlight recent work to raise standards says BGC [77]	*Casino Beats*	Industry press	15 March2021	NA	News report	Pro-industry
How to control gambling in a digital age [78]	*The Yorkshire Post*	Regional press	18 March 2021	Christopher Snowdon	Opinion piece	Pro-industry
How can we make gambling regulation fit for the digital age? [79]	*Politics Home*	Political press	31 March 2021	Jette Nygaard-Andersen, Chief Executive, Entain	Opinion piece	Pro-industry
Battles ahead for British bookies [80]	*iGaming Business*	Industry press	31 March 2021	Daniel O’Boyle	Review	Balanced
Striking the legal balance [81]	*Gambling Insider*	Industry press	19 April 2021	NA	Opinon piece	Balancd
Betting and gaming industry unites once again for safer gambling week 2021 [82]	*Politics Home*	Political press	20 April 2021	BGC	Opinion piece	Pro-industry
Betting and Gaming Council Statement—Response To The ‘Peers For Gambling Reform’ Report [83]	*Politics Home*	Political press	26 April 2021	BGC	Opinion piece	Pro-industry
Modernisation of Gambling Taxes and How It Will Affect the Industry [84]	*Warrington WorldWide*	Regional News	26 April 2021	Hannah Skentelbery	Review	Balanced
Gamcare urges a unified response to address gambling block loopholes [85]	*Casino Beats*	Industry press	26 April 2021	NA	News report	Balanced
Gamcare workshop highlights how gambling blocks can be bypassed [86]	*SBC News*	Industry news	27 April 2021	Ted Menmuir	News report	Pro-industry
Betting and Gaming Council’s response to peers for gambling reforms report [87]	*European Gaming*	Industry press	27 April 2021	Niji Narayan	News report	Pro-industry

## Data Availability

The data extraction database analysed during the current study are available from the corresponding author upon reasonable request.

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
