# Peer review of "Regulatory Resistance? Narratives and Uses of Evidence around “Black Market” Provision of Gambling during the British Gambling Act Review"

_ijerph, 2021, doi:10.3390/ijerph182111566_

Round 1

Reviewer 1 Report

  1. the authors have done a good job. however, I feel the paper lack innovativeness/newness.
  2. the introduction part seems a bit unorganized. I would suggest to re-write the introduction part. 
  3. it is suggested that the authors follow the PRISMA guideline for the paper. 
  4. the word  “black market” used in the title was not sufficiently covered in the paper in terms of peer-reviewed journals. thus, it is suggested that authors collect more reviews regarding this term from peer-reviewed journals.
  5. I would suggest including more peer-reviewed journals. A number of studies from 'trusted sources' should be enhanced. different search engines could be used for that. 

Author Response

Dear Editor/reviewers,

We thank the reviewers for their helpful comments on our manuscript, which we have taken on board and we believe have improved our paper. As the debate about policy change in Britain increases with pace, the industry narratives and framing documented in this paper are being accelerated and we believe it is vitally important to place focus on these processes in as close to real time as possible. To this end, we are heartened by the comments from Reviewer 2’s on the practical significance and innovation of our work.

In the sections that follow, we detail our response to each suggestion, noting action taken. We would be happy to discuss this further.

Best wishes,

Heather, Gerda, Fiona, Angela and Jeremy

Reviewer 1: comments

  1. the authors have done a good job. however, I feel the paper lack innovativeness/newness.

Response: We are pleased that this reviewer is positive about our work and approach. In terms of the innovation, we are unaware of any other published research that has systematically investigated gambling industry narratives and framing of issues in this way, especially as part of an organised campaign to influence policy change. Other contributions are mainly non-peer reviewed viewpoints and commentary (see Van Shalwyke et al 2021, for example). We believe it is vital to employ systematic, evidence-based, approaches to exploring how industry frame and thus attempt to shape policy decisions. We have made this clearer in our introduction.

  1. the introduction part seems a bit unorganized. I would suggest to re-write the introduction part. 

Response: in response to comments from this reviewer and suggestions from others, we have restructured the introduction.

  1. it is suggested that the authors follow the PRISMA guideline for the paper. 

Response: Where appropriate, we have presented the methods and the search according to PRISMA guidelines, see Figure 2, and, in accordance with best practice, submitted our pre-agreed search protocol as part of the supplementary material for this article, where more of this detail is documented. As this is paper is not a systematic review of evidence or a meta-analysis, but rather a qualitative and narrative a review of media articles, it means many of the items from the PRISMA checklist are not appropriate for this study. However, we have drawn on PRISMA guidelines, where possible, to add rigour and transparency to our processes, and increased these in response to Reviewer 2’s comments about heterogeneity.

  1. the word  “black market” used in the title was not sufficiently covered in the paper in terms of peer-reviewed journals. thus, it is suggested that authors collect more reviews regarding this term from peer-reviewed journals.

Response: In response to this comment and reviewer three, we have attempted to include more academic material on this topic. We note, however, there is a dearth of empirical examination on the “black market” with regard to gambling and have added a statement to that effect. Most articles are commentary or editorial pieces, or legal reviews from the perspective of moving from prohibition to legalisation, as a response to curtailing black market provision.

  1. I would suggest including more peer-reviewed journals. A number of studies from 'trusted sources' should be enhanced. different search engines could be used for that.

Response: This study seeks to examine the framing devices and narrative arguments put forward by industry around the “black market” in the early stages of the Gambling Act Review, and to do so in a way that is as close to real time as possible. Media is a key strategy used to influence public and political opinion and therefore it is appropriate that the focus of our empirical work is how industry narratives are rehearsed via media reports. This is not a review of academic literature on the black market for gambling. We agree that would be useful but is out of scope for this paper – which focuses on industry actions, following the mandate recommended by Knai et al.  

Reviewer 2 Report

Referee Report

Manuscript Title: Regulatory resistance: Narratives and uses of evidence around “black

market” provision of gambling during the British Gambling Act Review

Manuscript ID: ijerph-1400931

Journal: International Journal of Environmental Research and Public Health

Comments:

  1. I think the paper should take a neutral take on the gambling industry. Right in the abstract, the authors clearly convey to the readers that they have an unfavorable view of commercial gambling. They may have reasons to do so. However, as educated readers, we should be presented with facts on both sides and reach an informed decision on our own.
  2. The United States went through the prohibition of alcohol, which led to an increase in illegal alcohol production and consumption. It is not surprising that the gambling industry uses the black market argument. This makes perfect sense. The authors should acknowledge the relationship between the prohibition and the gambling, and produce arguments why the benefits outweigh the costs. Some examples:
  • https://prologue.blogs.archives.gov/2012/01/17/prohibition-and-the-rise-of-the-american-gangster/
  • https://www.theguardian.com/film/2012/aug/26/lawless-prohibition-gangsters-speakeasies
  • https://www.fbi.gov/history/brief-history/the-fbi-and-the-american-gangster
  1. The size of the current black market is a mute point. What matters is the expected increase in the size of the black market if a restriction is created. Here the authors should cite some historical evidence on the effect of restrictions/prohibitions on the black market. I like the evidence presented in 4. Discussion on the effect of gambling restrictions on the black market growth. I recommend that the authors include more examples of like those and present such evidence earlier in the paper (perhaps introduction) rather than the towards the end.
  2. I recommend the authors cite research on the legalization of medical and recreational use of marijuana in the US and how it affected the black market for this drug. For example:
  • A PBS article claiming that the legalization contributed to an increase in the black market (https://www.pbs.org/newshour/show/how-colorados-marijuana-legalization-strengthened-the-drugs-black-market)
  • A Politico article making a similar claim to the PBS article (https://www.politico.com/magazine/story/2019/07/21/legal-marijuana-black-market-227414/)
  • Research from Rutgers on the failed promise of legalization in reducing the black market activity (https://alcoholstudies.rutgers.edu/cannabis-black-market-thrives-despite-legalization/)
  • Research from Texas Tech University on the link between marijuana legalization and crime (https://papers.ssrn.com/sol3/papers.cfm?abstract_id=3271993)
  • NBER paper on the effect of legalization on the youth health outcomes (https://www.nber.org/system/files/working_papers/w23779/w23779.pdf)
  1. The paper cites news articles on the economic loss from regulating the gambling industry. The authors should go beyond reporting them. First of all, it makes sense that there will be economic loss (less gambling, hence less tax revenue). It behooves the authors to cite research on the improved health outcomes from further regulation. Perhaps, the savings from spending money on treating people with addiction problems outweigh the loss of tax revenue and employment. There are trade-offs, and the readers should be presented with the arguments on both sides.
  2. This is a general point. Why is gambling such a problem in the UK? If it is as bad as the authors claim, they should be more aggressive presenting evidence.

Author Response

Dear Editor/reviewers,

We thank the reviewers for their helpful comments on our manuscript, which we have taken on board and we believe have improved our paper. As the debate about policy change in Britain increases with pace, the industry narratives and framing documented in this paper are being accelerated and we believe it is vitally important to place focus on these processes in as close to real time as possible. To this end, we are heartened by the comments from Reviewer 2’s on the practical significance and innovation of our work.

In the sections that follow, we detail our response to each suggestion, noting action taken. We would be happy to discuss this further.

Best wishes,

Heather, Gerda, Fiona, Angela and Jeremy

Reviewer 2: Comments:

1. I think the paper should take a neutral take on the gambling industry. Right in the abstract, the authors clearly convey to the readers that they have an unfavorable view of commercial gambling. They may have reasons to do so. However, as educated readers, we should be presented with facts on both sides and reach an informed decision on our own.

Response: This was not our intention, and is likely an artefact of attempting to keep to the word limit. We have amended the opening sentence slightly to reflect where thinking is going on this issue.

2. The United States went through the prohibition of alcohol, which led to an increase in illegal alcohol production and consumption. It is not surprising that the gambling industry uses the black market argument. This makes perfect sense. The authors should acknowledge the relationship between the prohibition and the gambling, and produce arguments why the benefits outweigh the costs. Some examples:

  • https://prologue.blogs.archives.gov/2012/01/17/prohibition-and-the-rise-of-the-american-gangster/
  • https://www.theguardian.com/film/2012/aug/26/lawless-prohibition-gangsters-speakeasies
  • https://www.fbi.gov/history/brief-history/the-fbi-and-the-american-gangster

Response: Thanks for these helpful documents; we have added more on this in the discussion and the introduction, including the point that there isn’t necessarily a dichotomy between prohibition and licensing, but rather the shade and degree of regulatory approach that exist in between both.

3. The size of the current black market is a moot  point. What matters is the expected increase in the size of the black market if a restriction is created. Here the authors should cite some historical evidence on the effect of restrictions/prohibitions on the black market. I like the evidence presented in 4. Discussion on the effect of gambling restrictions on the black market growth. I recommend that the authors include more examples of like those and present such evidence earlier in the paper (perhaps introduction) rather than the towards the end.

Response: we agree that the expected size of the increase is of most importance. However, very few jurisdictions have imposed tighter regulation upon a pre-existing, legal, gambling market and those which have (including Britain) have not evaluated the impact of these movements. The evidence base is therefore nascent, which allows space for the gambling industry to fill this void. As suggested, we have moved discussion of what evidence we have to the introduction and added observations to this effect to the introduction also.

4. I recommend the authors cite research on the legalization of medical and recreational use of marijuana in the US and how it affected the black market for this drug. For example:

  • A PBS article claiming that the legalization contributed to an increase in the black market (https://www.pbs.org/newshour/show/how-colorados-marijuana-legalization-strengthened-the-drugs-black-market)
  • A Politico article making a similar claim to the PBS article (https://www.politico.com/magazine/story/2019/07/21/legal-marijuana-black-market-227414/)
  • Research from Rutgers on the failed promise of legalization in reducing the black market activity (https://alcoholstudies.rutgers.edu/cannabis-black-market-thrives-despite-legalization/)
  • Research from Texas Tech University on the link between marijuana legalization and crime (https://papers.ssrn.com/sol3/papers.cfm?abstract_id=3271993)
  • NBER paper on the effect of legalization on the youth health outcomes (https://www.nber.org/system/files/working_papers/w23779/w23779.pdf)

Response: we thank the reviewer for these suggestions. It is certainly useful to be able to draw out some of these parallels. it is very interesting to note these non-linear associations, given the expressed assumption movements from prohibition to legalisation will diminish the black market, we have added short sentence on this and increased our discussion of the factors that are likely to influence the shape for this relationship for gambling.  We have also increased our comparisons with the evidence base for tobacco, as gambling, like tobacco is something that is currently legal but has faced greater restrictions – it is therefore helpful to see parallels with other sectors.

5. The paper cites news articles on the economic loss from regulating the gambling industry. The authors should go beyond reporting them. First of all, it makes sense that there will be economic loss (less gambling, hence less tax revenue). It behooves the authors to cite research on the improved health outcomes from further regulation. Perhaps, the savings from spending money on treating people with addiction problems outweigh the loss of tax revenue and employment. There are trade-offs, and the readers should be presented with the arguments on both sides.

Response: Thanks for this. We have added evidence from two independent economic reports which both argued that restricting gambling would have a net benefit for the economy, and also cited recent research on the social costs of gambling harms.

6. This is a general point. Why is gambling such a problem in the UK? If it is as bad as the authors claim, they should be more aggressive presenting evidence.

Response: we have added notes about the social cost of gambling harms to both the introduction and the discussion.

Reviewer 3 Report

The regulation of the gambling industry has the value and significance of discussion in the global scope,the author takes the media’s discussion and analysis of "black market" behavior in the gaming industry as a reference for policy actions and regulatory resistance, which is of practical significance and innovative perspective. However, this article has the following serious problems, which would be further considered by the author:

  • The title emphasizes "regulation", but the article is more about the influence to the whole Gambling Act. The act is wide-ranging including regulation of lotteries、licenses、the protection of Children, etc. But this article only focus on “Black market”. Authors need to consider adjusting the title, or limit the content of the literature to the regulatory part of “Black market”, to emphases the influencing factors media articles mentions, instead of all.
  • Table 1 presents 55 articles after screening. However, analysis based on media articles has two objective defects: subjectivity,and the consistency of the opinion of mainstream media. I noticed that the author did not perform relevant autocorrelation and heterogeneity analysis after screening these articles. This will lead to unstable and biased conclusions.
  • The conclusion is more intuitive and has strong limitations. The author mentioned in the conclusion that “To support these debates, understanding of the “black market” in Britain needs to be supported by credible evidence based on independent and rigorous research.”, As the author said, the conclusion analysis of this article still lacks realistic references value.

Author Response

Dear Editor/reviewers,

We thank the reviewers for their helpful comments on our manuscript, which we have taken on board and we believe have improved our paper. As the debate about policy change in Britain increases with pace, the industry narratives and framing documented in this paper are being accelerated and we believe it is vitally important to place focus on these processes in as close to real time as possible. To this end, we are heartened by the comments from Reviewer 3’s on the practical significance and innovation of our work.

In the sections that follow, we detail our response to each suggestion, noting action taken. We would be happy to discuss this further.

Best wishes,

Heather, Gerda, Fiona, Angela and Jeremy

Reviewer 3: Comments

The regulation of the gambling industry has the value and significance of discussion in the global scopethe author takes the media’s discussion and analysis of "black market" behavior in the gaming industry as a reference for policy actions and regulatory resistance, which is of practical significance and innovative perspective. However, this article has the following serious problems, which would be further considered by the author:

  • The title emphasizes "regulation", but the article is more about the influence to the whole Gambling Act. The act is wide-ranging including regulation of lotteries、licenses、the protection of Children, etc. But this article only focus on “Black market”. Authors need to consider adjusting the title, or limit the content of the literature to the regulatory part of “Black market”, to emphases the influencing factors media articles mentions, instead of all.

Response: Thanks for this point. We see the industry actions around the Black Market as one part of their multi-pronged strategy to enact Regulatory Resistance in the wake of the Gambling Act Review, and therefore our preference is to retain this. We have increased discussion of the way the debate is being framed, with greater regulation having a direct relationship with black market size in our introduction.

  • Table 1 presents 55 articles after screening. However, analysis based on media articles has two objective defects: subjectivity,and the consistency of the opinion of mainstream media. I noticed that the author did not perform relevant autocorrelation and heterogeneity analysis after screening these articles. This will lead to unstable and biased conclusions.

Response: This review is a narrative, qualitative synthesis of media articles focusing on identifying key discursive themes about how debates about the black market are being shaped. As such, it is situated within the interpretative paradigm which is not really amenable to the types of analyses mentioned here. That said, understanding the heterogeneity of the articles is important and we have added more detail into our methods section and on the characteristics of the studies to assess this. In this instance, we believe the variability of articles serves to enhance rather than undermine our results: we have made clear that themes presented do not just cut across those studies published by industry press, but are also evident within those published by mainstream media. This was already reflected in our discussion. We have added more information to Table 1 about our coding for the tone of each article also to make the variability across the review articles clearer. Our approach is guided by PRISMA reporting guidelines which state ; “If other methods were used to explore heterogeneity because data were not amenable to meta-analysis of effect estimates, describe the methods used (such as structuring tables to examine variation in results across studies based on subpopulation, key intervention components, or contextual factors). (See here: https://www.bmj.com/content/372/bmj.n160 ). This is what we have done by looking at variation among article types based on their publication source.

  • The conclusion is more intuitive and has strong limitations. The author mentioned in the conclusion that “To support these debates, understanding of the “black market” in Britain needs to be supported by credible evidence based on independent and rigorous research.”, As the author said, the conclusion analysis of this article still lacks realistic references value.

Response; we have strengthened the conclusions to say more about the lack of credible evidence available, thus creating space to allow industry-funded research and narratives to fill this gap and, potentially, capture the debate. This has been seen in tobacco research also. We have also noted that years of systemic under-funding in gambling research in Britain, especially when regulatory change has been implemented, further exacerbates this situation – as prime opportunities to generate useful evidence on this have been missed. We have added some of this text to the introduction also to bring this together. We believe it is important to draw attention to these key issues.

Round 2

Reviewer 1 Report

thank you for replying to all the stated concerns. however, the authors may think about re-constructing the paper, especially section 3(result). I believe that would make the paper easy for readers( seems a bit more sub-sections). 

additionally, pls change the type of article from "research article" to "review article". 

Author Response

We thank the reviewer for their comments. We have added a bit more signposting at Section 3.2 about the results, and changed the numbered sub-sections so that the themes flow a bit clearer in this section.

We are happy to change the article type to a review, and will speak to the Editors about this.

Reviewer 2 Report

The paper is significantly improved. I am in favor of accepting it in its present form.

Author Response

Many thanks for your useful comments, which significantly improved this manuscript.

Reviewer 3 Report

I noticed that the authors took my suggestions and revised this article. The responses of the author are well-founded, explaining and accepting the questions and suggestions I put forward in the previous review of the article, and modifying the main content of the article according to relevant suggestions, and the abstract and conclusion are more in line with the ideas expressed in the title of the article.To sum up, I think this article meets the requirements for publication and agree to publish it.It is hoped that the author can conduct more in-depth research in this field, add data and empirical analysis, and make the results more delicate. Wish them luck.

Author Response

Many thanks for your exceptionally helpful comments which have improved our manuscript. Please do get in touch with us if you are interested in knowing more about work in this area. 

Heather